# Boosting Multi-Domain Fine-Tuning of Large Language Models through Evolving Interactions between Samples

Xize Liang[†][1]   Lin Yang[2]   Jie Wang[1]   Yiyang Lu[1]   Runyu Wu[1]   Hanzhu Chen[1]   Jianye Hao[2][3]

## Abstract

The multi-domain fine-tuning of large language models (LLMs) confronts a notorious trade-off among abilities across domains. Existing studies attribute this trade-off to the conflicts between samples rooted in inherent semantics. Recent approaches attempt to mitigate these conflicts through the empirical investigation or heuristic strategies. However, without a fundamental understanding of interactions between samples, they yield only marginal improvements, while incurring substantial trial-and-error costs. To address this challenge, we move beyond empirical studies by modeling interactions between samples as their influence on each other's loss, estimated using gradients. Intriguingly, we find that these interactions **evolve throughout training** rather than being purely determined by inherent semantics. Building on this insight, we propose __EV__olving __I__nteraction-guided __C__urriculum (**EVIC**), which iteratively selects samples that positively influence the overall dataset for training. By dynamically adapting the training curriculum to prioritize samples that contribute the most to the model training, EVIC effectively mitigates conflicts and improves the performance-to-sample ratio. Extensive experiments on a mixed dataset covering coding, math, and general tasks with several model architectures show that EVIC significantly outperforms all baselines across diverse capabilities.

## 1. Introduction

Large language models (LLMs) have demonstrated remarkable capabilities in various fields, such as general instruction following (Yang et al., 2024b; Xu et al., 2024a; Dong et al., 2024), code generation (Jiang et al., 2024; Hou et al., 2024; Hui et al., 2024), and mathematical reasoning (Ahn et al., 2024; Imani et al., 2023; Luo et al., 2023). Building on these successes, cutting-edge LLMs are striving to integrate diverse capabilities across various domains, aiming to better address read-world tasks and advance the realization of general artificial intelligence (Dubey et al., 2024; Yang et al., 2024a; Liu et al., 2024a; Adler et al., 2024).

However, the multi-domain fine-tuning of LLMs still confronts significant challenges. In particular, training a model that excels across all domains is extremely difficult, as improvements in one capability often come at the expense of others (Wang et al., 2023a; Leybzon & Kervadec, 2024; Mueller et al., 2022). Existing studies attribute this issue to inherent conflicts between samples, arising from their different semantics (Wu et al., 2024; Dong et al., 2023; Wang et al., 2023b; Ge et al., 2023). A common alternative is to train specialized models for each domain and then integrate them (Feng et al., 2024; Zhou et al., 2024; Wang et al., 2024), but this inevitably incurs considerable computational costs. Furthermore, from the perspective of data resources, conflicts between samples hinder the effective use of high-quality data, which is especially valuable due to its scarcity and high acquisition costs (Li et al., 2023a; Liu et al., 2024b; Li et al., 2023b). Therefore, it is highly desirable to explore strategies to fully leverage mixed datasets for the enhancement of LLMs' multi-domain capabilities.

Existing efforts for this problem rely on empirical investigation or heuristic strategies to manage training samples. Based on empirical insights gained from experiments on a mixed dataset containing samples from mathematics, code, and general domains, Dong et al. (2023) propose Dual-stage Mixed Fine-tuning (DMT), which trains on full amounts of specialized (i.e., math and code) data first and then trains on general data with a small amount of specialized data included. Such experience-based approaches may be useful in certain scenarios but tend to be inefficient, requiring numerous experiments to discover suitable training strategies.

Work done during Xize Liang 's research internship at Huawei. [†]<xizeliang@mail.ustc.edu.cn> [1]MoE Key Laboratory of Brain-inspired Intelligent Perception and Cognition, University of Science and Technology of China [2]Noah's Ark Lab, Huawei Technologies [3]College of Intelligence and Computing, Tianjin University. Correspondence to: Jie Wang <jiewangx@ustc.edu.cn>.

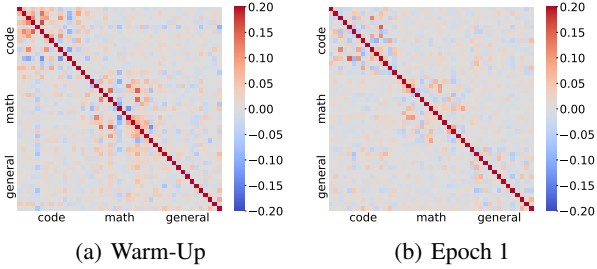

(a) Warm-Up       (b) Epoch 1

*Figure 1.* The interaction matrices computed after training Mistral-7B for warm-up and one epoch. We randomly select 15 samples from each domain to generate the heatmap. The indices in the figure, ordered from smallest to largest, correspond to samples from coding, math, and general domains. We divide the elements of both matrices by $10^8$ and clip them to the range of $[-0.2, 0.2]$.

Alternatively, Mixture-of-Skills (MoS) employs reward signals based on heuristic metrics such as transferability and difficulty to train a scorer network, which is then used to assign sampling probabilities to training samples. Compared to experience-driven methods, heuristic-based methods offer more guided curriculum strategies but also lead to only marginal improvements due to the lack of a fundamental understanding of interactions between samples.

To address this challenge, we move beyond empirical studies and propose a novel LLM fine-tuning framework named **EV**olving **I**nteraction-guided **C**urriculum (**EVIC**). Specifically, we model the interactions between samples as their influence on each other's loss, estimated using (Adam) gradients. Notably, we find that these interactions **evolve during the training process** (as shown in Figure 1), rather than being purely determined by inherent semantics. Inspired by this intriguing observation, EVIC periodically updates the interactions between each pair of samples and selects the samples that positively influence the overall dataset for training. By dynamically adapting the training curriculum to prioritize samples that contribute the most to the model training, EVIC effectively mitigates conflicts and significantly improves the *performance-to-sample ratio*. That is, given the sample batch size, EVIC achieves higher performance with fewer training steps compared to existing methods.

The main contributions of this paper are as follows.

- We find that the interactions between samples estimated with (Adam) gradients evolve during the training process, providing insights into enhancing LLM fine-tuning on multi-domain datasets through progressive data management.

- We propose the EVIC framework, which effectively mitigates conflicts between samples and achieves superior performance-to-sample ratio by dynamically adapt-

ing the training curriculum to prioritize samples that contribute the most to the model training.

- We conduct extensive experiments on a mixed dataset containing 182,166 samples covering the domains of mathematical reasoning, code generation, and general instruction following with Mistral-7B, Llama-3.1-8B, and Qwen2.5-14B. The evaluation results on GSM8K, HumanEval, and AlpacaEval 2.0 show that EVIC outperforms all baselines across diverse capabilities.

## 2. Preliminaries

**Supervised Fine-Tuning (SFT).** An LLM $\pi_\theta$ (with parameters $\theta$) generates a response $\mathbf{y} = [y_1, \ldots, y_m]$ to the query $\mathbf{x} = [x_1, \ldots, x_n]$, where the tokens $(x_i)_{i=1}^n$ and $(y_j)_{j=1}^m$ come from a predefined vocabulary, in an autoregressive paradigm. Specifically, the model samples $y_j$ from the conditional probability distribution $\pi_\theta(\cdot \mid \mathbf{x}, \mathbf{y}_{1:j-1})$, where $\mathbf{y}_{1:0}$ is null and $\mathbf{y}_{1:j-1} = [y_1, \ldots, y_{j-1}]$ for $j = 2, \ldots, m$. We can decompose the conditional probability $\pi_\theta(\mathbf{y} \mid \mathbf{x})$ into $\pi_\theta(\mathbf{y} \mid \mathbf{x}) = \prod_{j=1}^m \pi_\theta(y_j \mid \mathbf{x}, \mathbf{y}_{1:j-1})$. We fine-tune the model on a dataset $D = \{(\mathbf{x}^{(i)}, \mathbf{y}^{(i)})\}_{i=1}^N$ with the loss function $\mathcal{L}(\theta; D) = \frac{1}{N} \sum_{i=1}^N \ell(\theta; \mathbf{x}^{(i)}, \mathbf{y}^{(i)}) = -\frac{1}{N} \sum_{i=1}^N \log \pi_\theta(\mathbf{y}^{(i)} \mid \mathbf{x}^{(i)})$.

**Vanilla Curriculum Learning (CL).** The key idea of vanilla CL is to train the model on a sequence of tasks or samples that gradually increase in difficulty (Bengio et al., 2009; Soviany et al., 2022). The model is initially presented with simpler samples, allowing it to learn fundamental concepts before moving on to more complex tasks. Formally, for the dataset $D = \{(\mathbf{x}^{(i)}, \mathbf{y}^{(i)})\}_{i=1}^N$, a curriculum consists of a sequence of subsets $(D_k)_{k=1}^K$, where $D_k = \{\mathbf{x}^{(i_j^k)}, \mathbf{y}^{(i_j^k)}\}_{j=1}^{N_k}$ and the difficulty increases with $k$. Then, the model learns $(D_k)_{k=1}^K$ in sequence. The vanilla CL approaches use various metrics to measure difficulty, such as sample complexity, model uncertainty, or domain-specific heuristics.

**Fine-Tuning with Adam.** The Adam optimizer (Kingma, 2014) is widely used for fine-tuning LLMs, which combines the advantages of both adaptive gradient algorithms and momentum methods. Starting from the initial parameter $\theta_0$, the update rule of the Adam optimizer is

$$\theta_t = \theta_{t-1} - \eta_{t-1}\Gamma(\theta_{t-1}),$$
$$\Gamma(\theta_{t-1}) = \frac{\mathbf{m}_t}{\sqrt{\mathbf{v}_t} + \varepsilon}$$

for $t \geq 1$, where

$$\mathbf{m}_t = \begin{cases} \frac{1}{1-\beta_1^t}(\beta_1 \mathbf{m}_{t-1} + (1-\beta_1)\nabla_\theta \mathcal{L}(\theta_{t-1})), & (t \geq 1) \\ \mathbf{0}, & (t = 0) \end{cases}$$

and

$$\mathbf{v}_t = \begin{cases} \frac{1}{1-\beta_2^t}(\beta_2 \mathbf{v}_{t-1} + (1-\beta_2)\nabla_\theta \mathcal{L}(\theta_{t-1})), & (t \geq 1) \\ \mathbf{0}, & (t = 0) \end{cases}$$

represent the first and second moment estimates, respectively, and $\beta_1, \beta_2$ are hyperparameters that control the exponential decay rates of these moment estimates. Here, we follow the notation $\Gamma$ from (Xia et al., 2024a) to denote the Adam gradient. We omit the batch of samples $B_t$ in $\nabla_\theta \mathcal{L}(\theta_{t-1}; B_t)$ and $\Gamma(\theta_{t-1}; B_t)$ for the simplicity of notation, unless we want to emphasize $B_t$.

# 3. Evolving Interaction-guided Curriculum

We propose **EV**olving **I**nteraction-guided **C**urriculum (**EVIC**), a novel approach with minimal manual design cost, sample-level data management, and superior performance-to-sample ratio. Specifically, we start from the perspective of learning dynamics and model the inter-sample interactions as their influence on each other's loss, estimated using (Adam) gradients (Section 3.1). Then, we observe that these interactions evolve throughout the training process, rather than being purely determined by intrinsic semantics (Section 3.2). Building on this insight, we periodically update the estimation of inter-sample interactions during training and select samples that positively interact with the overall dataset for subsequent training (Section 3.3).

## 3.1. Modeling Interactions between Samples

The learning dynamics provides an effective way to understand and quantify how samples influence each other during the training process. Inspired by (Xia et al., 2024a), which uses loss-based dynamics to estimate the similarity between training and test samples, we model the interactions between training samples as their influence on each other's loss. Consider two samples $s^{(i)} = (\mathbf{x}^{(i)}, \mathbf{y}^{(i)})$ and $s^{(j)} = (\mathbf{x}^{(j)}, \mathbf{y}^{(j)})$, the influence of $s^{(j)}$ on the loss of $s^{(i)}$ at parameter $\theta_t$ can be estimated by the Taylor expansion as

$$\ell\left(\theta_{t+1}^{(j)}; \mathbf{x}^{(i)}, \mathbf{y}^{(i)}\right) - \ell\left(\theta_t; \mathbf{x}^{(i)}, \mathbf{y}^{(i)}\right)$$
$$\approx \nabla_\theta \ell\left(\theta_t; \mathbf{x}^{(i)}, \mathbf{y}^{(i)}\right)^\top \cdot \left(\theta_{t+1}^{(j)} - \theta_t\right)$$
$$= -\eta_t \left\langle \underbrace{\nabla_\theta \ell\left(\theta_t; \mathbf{x}^{(i)}, \mathbf{y}^{(i)}\right)}_{\text{gradient of } s^{(i)}}, \underbrace{\Gamma\left(\theta_t; \mathbf{x}^{(j)}, \mathbf{y}^{(j)}\right)}_{\text{Adam gradient of } s^{(j)}} \right\rangle, \quad (1)$$

where $\theta_{t+1}^{(j)}$ is the parameter after learning $s^{(j)}$ from $\theta_t$ and $\eta_t$ is the learning rate at the $t$-th step.

**Random gradient projection.** For LLMs, the high dimensionality of (Adam) gradients imposes a prohibitive computational overload on computing Eq. (1). Following

(Xia et al., 2024a), we use the Johnson-Lindenstrauss (JL) transformation to project the (Adam) gradients into an 8192-dimensional space, which preserves inner products within an acceptable margin of error (see Appendix C.1 for details). Formally, we denote an (Adam) gradient $\mathbf{g}$ after projection as $\hat{\mathbf{g}} \in \mathbb{R}^{8192}$. Then, for the dataset $D = \{(\mathbf{x}^{(i)}, \mathbf{y}^{(i)})\}_{i=1}^N$, we use an *interaction matrix* $\mathbf{Int}(\theta_t) \in \mathbb{R}^{N \times N}$ to represent the interactions between samples, where

$$\mathbf{Int}(\theta_t)[j, i] = \left\langle \underbrace{\widehat{\nabla_\theta \ell}\left(\theta_t; \mathbf{x}^{(i)}, \mathbf{y}^{(i)}\right)}_{\substack{\text{projected} \\ \text{gradient of } s^{(i)}}}, \underbrace{\widehat{\Gamma}\left(\theta_t; \mathbf{x}^{(j)}, \mathbf{y}^{(j)}\right)}_{\substack{\text{projected Adam} \\ \text{gradient of } s^{(j)}}} \right\rangle,$$

$$(2)$$

where $\mathbf{Int}(\theta_t)[j, i] \geq 0$ indicates that $s_j$ promotes the learning of $s_i$, while $\mathbf{Int}(\theta_t)[j, i] < 0$ indicates the opposite. By the linearity of the inner product, the estimated influence of $s_j$ on the overall dataset is given by the $j$-th row sum of $\mathbf{Int}(\theta_t)$, i.e., $\sum_{i=1}^N \mathbf{Int}(\theta_t)[j, i]$.

## 3.2. Observations of Inter-Sample Interactions

**Assumptions underlying existing approaches.** Studies based on learning dynamics typically compute gradients and related quantities (e.g., sample importance) using early-stage checkpoints in the training process (Xia et al., 2024a; Pruthi et al., 2020). Research on fine-tuning LLMs on multi-domain datasets, such as (Dong et al., 2023), generally treats all samples from the same domain as a whole, focusing on data management at the domain granularity. These practices imply two underlying assumptions as follows.

1. *Inter-sample interactions remain **largely unchanged** throughout the training process.*

2. *Samples within the **same domain mutually enhance** each other's learning.*

However, our observations suggest that **these two customary assumptions do not always hold**. On a mixed dataset composed of CodeAlpaca, GSM8K-RFT, and Alpaca-GPT4 (the three datasets correspond to domains of code generation, mathematical reasoning, and general instruction following, respectively. See Section 4.1 for details), we first randomly select 5% of the samples and perform warm-up training for Llama-3.1-8B to adapt it from the pre-training distribution to the SFT distribution. After that, we conduct one epoch of training on the entire dataset. We compute the interaction matrices as in Eq. (2) and randomly sample 15 samples from each domain to plot the interaction heatmap, as shown in Figure 1. From the heatmap, we make the following interesting observations.

1. The signs and absolute values of the interactions between many pairs of samples change significantly dur-

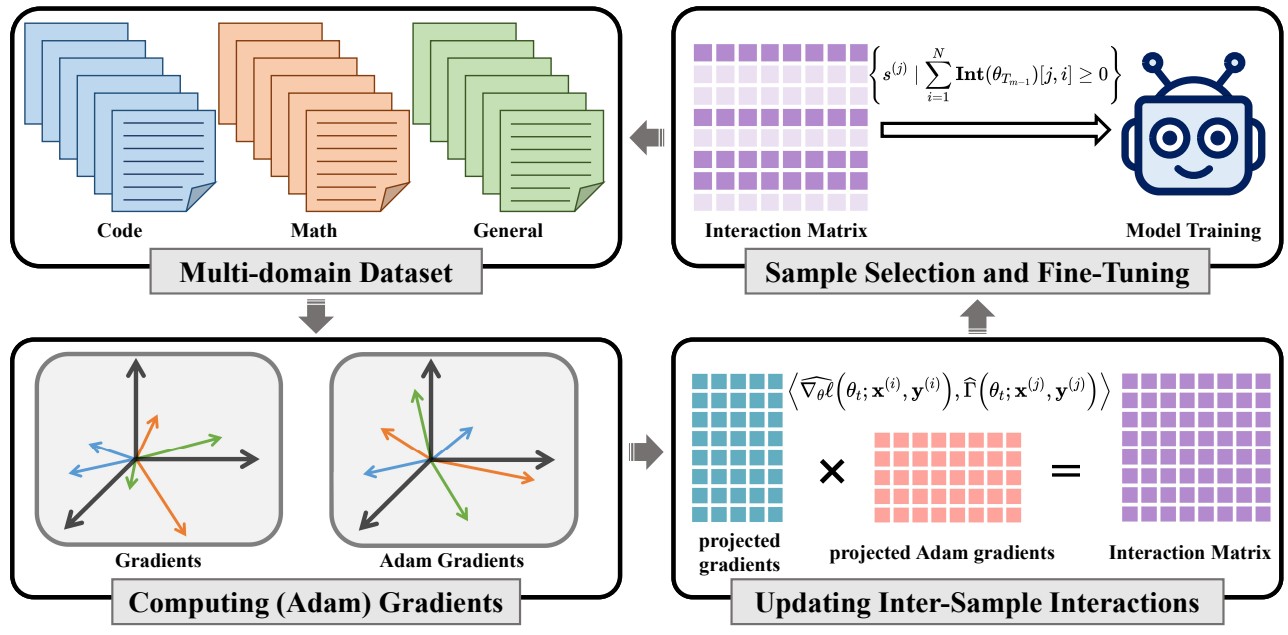

**Figure 2.** Framework of EVIC. Given a multi-domain mixed dataset, EVIC first obtain (Adam) gradients for the entire dataset in each stage. Then, EVIC computes the inter-sample interaction matrix using the (Adam) gradients. After that EVIC selects samples that positively interact with the entire dataset for training.

ing training, indicating that the interactions evolve dynamically over the course of training.

2. For some sample pairs $(s_i, s_j)$, $\text{sign}\left(\textbf{Int}(\theta_t)[j, i]\right)$ is not necessarily equal to $\text{sign}(\textbf{Int}\,(\theta_t)[i, j])$, which means that the interaction relationship are asymmetric.

3. Conflicts also exist among samples within the same domain, while some samples from different domains mutually promote each other's learning, suggesting that managing samples at the sample-level may be more effective than at the domain-level.

### 3.3. Dynamic Interaction-guided Curriculum Learning

The above intriguing observations inspire us to select samples that have a positive influence on the overall dataset in a staged manner during training, thus leveraging the dynamic nature of interactions between samples. We summarize the EVIC framework in Algorithm 1. Specifically, we first randomly select 5% of the samples (for the ablation study on the proportion, please see Section 4.3) from the multi-domain dataset for warm-up training (Lines 2-5 in Algorithm 1). Then, we compute the interaction matrix for the entire dataset (Lines 7-9 in Algorithm 1) and select samples corresponding to non-negative-sum rows for training (Lines 10-14 in Algorithm 1). After that, we update the interaction matrix, select samples, train the model, and repeat this process until the end of the training process.

---

**Algorithm 1** The EVIC framework

1: **Input:** dataset $D = \{s^{(i)}\}_{i=1}^{N}$, number of iterations $M$, base model $\pi_{\text{base}}$
2: **Warm-Up and Initialization:**
3:     $D_0 \leftarrow$ Randomly selected 5% of samples from $D$
4:     Train $\pi_{\text{base}}$ on $D_0$ to form $\pi_{\theta_0}$
5:     $T_0 \leftarrow 0$
6: **for** $m = 1, \ldots, M$ **do**
7:     ▶ **Interaction Matrix Computation:**
8:         Compute JL-projected (Adam) gradients for $D$
9:         Compute $\textbf{Int}(\theta_{T_{m-1}})$ according to Eq. (2)
10:    ▶ **Sample Selection:**
11:      $D_m \leftarrow \left\{ s^{(j)} \mid \sum_{i=1}^{N} \textbf{Int}(\theta_{T_{m-1}})[j, i] \geq 0 \right\}$
12:      Shuffle $D_m$
13:    ▶ **Model Training:**
14:      $\pi_{\theta_{T_m}} \leftarrow$ fine-tune $\pi_{\theta_{T_{m-1}}}$ on $D_m$.
15: **end for**
16: **Output:** $\pi_{\theta_{T_M}}$

---

**Discussion.** Although our framework does not impose a constraint requiring every sample to be selected, our experiments show that nearly all samples are learned at least once. We provide an intuitive explanation for this, and leave a more rigorous analysis of this phenomenon for our future work. First, because many inter-sample interactions change their signs during the training process, pairs of sam-

Table 1. **Statistics of the datasets and benchmarks for each domain, and settings for model inference and evaluation.**

| Domain | $D_{\text{train}}$ | $|D_{\text{train}}|$ | $D_{\text{test}}$ | $|D_{\text{test}}|$ | Temp | Seed |
|---|---|---|---|---|---|---|
| Code | CodeAlpaca | 20,022 | HumanEval | 164 | 0.3 | 1-10 |
| Math | GSM8K-RFT | 110,142 | GSM8K-test | 1,319 | 0.0 | - |
| General | Alpaca-GPT4 | 52,002 | AlpacaEval 2.0 | 805 | 0.7 | 1-10 |

ples that are initially in conflict may gradually become less conflicting or even mutually promoting as the training progresses. Second, empirically, gradients at parameters near local optima tend to have smaller magnitudes. Thus, even if a pair of samples remains in conflict throughout training, the absolute value of their interaction will diminish as one of the samples is learned. This causes the contribution of these conflicting pairs to the overall influence on the dataset to decrease, thereby increasing the likelihood of selecting samples that have not yet been learned.

## 4. Experiments

### 4.1. Experimental Settings

**Tasks and Datasets.** We focus on improving the comprehensive capabilities of LLMs in code generation, mathematical reasoning, and general instruction following, which are the three most attention-grabbing abilities of LLMs at present. We use three datasets for each domain: CodeAlpaca (Chaudhary, 2023) for code generation, GSM8K-RFT (Cobbe et al., 2021; Yuan et al., 2023) for mathematical reasoning, and Alpaca-GPT4 (Peng et al., 2023) for general instruction following. For details of the datasets, see Appendix A.1.

**Evaluation.** We evaluate the performance of code generation, mathematical reasoning, and general instruction following of all models on HumanEval (Chen et al., 2021), GSM8K-test (Cobbe et al., 2021), and AlpacaEval 2.0 (Dubois et al., 2024), respectively. The statistics of the datasets and benchmarks for each domain, and the settings for model inference and evaluation are presented in Table 1.

Specifically, for HumanEval, we perform model inference with a temperature of 0.3 and random seeds ranging from 1 to 10, and report the mean and standard deviation of Pass@1 and Pass@10 of all models. For GSM8K-test, we employ the greedy decoding with a temperature of 0.0 and report the accuracies. For AlpacaEval 2.0, we perform model inference with a temperature of 0.7 and random seeds ranging from 1 to 10. We employ GPT-4—the default evaluator of AlpacaEval—as the judge to compare the outputs of all models with those of GPT-4-Turbo, using the length-controlled win rate and the win rate as metrics. In addition to the individual metrics for each domain, we also report their average.

Specifically, we compute the average metric by

$$\text{AVG} = \frac{1}{3}\left(\text{ACC} + \frac{\text{P@1} + \text{P@10}}{2} + \frac{\text{LC-WR} + \text{WR}}{2}\right).$$

We use vLLM (Kwon et al., 2023) for efficient model inference. For more details, please see Appendix A.4.

**Baselines and Hyperparameters.** We compare our proposed EVIC framework with the baselines as follows.

- **Multi-Task Learning** (**MTL**) trains on the multi-domain dataset without any sample management strategies. MTL belongs to *progressive CL* methods as categorized in (Soviany et al., 2022).

- **Domain-Specific Learning** (**DSL**) trains only on the separate dataset for each domain, which reflects the upper limit of the model's specialized ability on each dataset. We use $\text{DSL}_{\text{code}}$, $\text{DSL}_{\text{math}}$, and $\text{DSL}_{\text{gen}}$ to represent the versions of DSL training only on code, math, and general samples, respectively.

- **Dual-stage Mixed Fine-tuning** (**DMT**, Dong et al. (2023)) trains on full amounts of specialized (i.e., code and math) samples first and then trains on general-domain samples with a small proportion of the specialized samples included.

- **Mixture-of-Skills** (**MoS**, Wu et al. (2024)) employs reinforcement learning with reward signals based on heuristic metrics such as transferability and difficulty to train a scorer network, and then use it to assign sampling probabilities to samples. MoS belongs to *balanced CL* methods as categorized in (Soviany et al., 2022).

Besides, we also add *vanilla CL* and *self-pased CL* as our additional baselines in Appendix B.2.

**Model Training.** We conduct all the model training *once* with LoRA (Hu et al., 2021) implemented in the LLaMA-Factory framework (Zheng et al., 2024). We run DSL until its test performance no longer improves. For MTL, DMT, and MoS, we align their training process to three epochs, as their performance shows only marginal gains beyond three epochs. Because EVIC selects different samples at each stage, we measure its training duration in terms of *iterations*[1] and *training steps* (with a batch size of 128), and ensure that **the total training steps of EVIC do not exceed those of the baseline methods** to maintain fairness. We run all experiments on 8 NVIDIA A100 GPUs (80GB). For more details of our baselines, hyperparameters, and the training process, see Appendix A.2.

---

[1]An iteration refers to performing one sample selection phase and training the model on the selected samples for one epoch.

*Table 2.* **Performance of models trained on the multi-domain dataset with different methods, evaluated on multiple benchmarks.** The bold font indicates the best result and an underline indicates the second-best result. Please note that **DSL is not involved in the performance comparison** and only serves as a reference for the upper limit of each ability.

| Model | Mistral-7B-v0.3 | | | | | | Llama-3.1-8B | | | | | | Qwen2.5-14B | | | | | |
|---|---|---|---|---|---|---|---|---|---|---|---|---|---|---|---|---|---|---|
| $D_{\text{test}}$ | GSM8K | HumanEval | | AlpacaEval 2 | | | GSM8K | HumanEval | | AlpacaEval 2 | | | GSM8K | HumanEval | | AlpacaEval 2 | | |
| Metric | ACC | P@1 | P@10 | LC-WR | WR | *AVG* | ACC | P@1 | P@10 | LC-WR | WR | *AVG* | ACC | P@1 | P@10 | LC-WR | WR | *AVG* |
| $DSL_{\text{math}}$ | 59.1 | $0.2_{\pm0.1}$ | $1.4_{\pm0.4}$ | $0.6_{\pm0.2}$ | $0.7_{\pm0.2}$ | *20.2* | 68.3 | $6.7_{\pm0.2}$ | $19.0_{\pm1.2}$ | $0.9_{\pm0.2}$ | $1.1_{\pm0.2}$ | *27.4* | 85.0 | $57.5_{\pm0.5}$ | $81.0_{\pm1.5}$ | $3.8_{\pm0.5}$ | $3.3_{\pm0.3}$ | *52.6* |
| $DSL_{\text{code}}$ | 0.1 | $40.5_{\pm0.7}$ | $58.5_{\pm1.4}$ | $7.5_{\pm0.7}$ | $3.8_{\pm0.3}$ | *18.4* | 13.7 | $48.8_{\pm0.5}$ | $71.2_{\pm0.4}$ | $8.1_{\pm0.6}$ | $3.9_{\pm0.2}$ | *26.6* | 60.4 | $71.6_{\pm0.5}$ | $89.2_{\pm1.4}$ | $10.6_{\pm1.0}$ | $4.8_{\pm0.4}$ | *49.5* |
| $DSL_{\text{gen}}$ | 37.6 | $11.6_{\pm0.6}$ | $33.9_{\pm2.3}$ | $11.6_{\pm1.3}$ | $7.0_{\pm0.4}$ | *23.2* | 44.1 | $6.9_{\pm0.4}$ | $23.0_{\pm2.1}$ | $12.5_{\pm1.2}$ | $6.6_{\pm0.5}$ | *22.9* | 73.6 | $3.1_{\pm0.2}$ | $13.4_{\pm1.5}$ | $19.3_{\pm1.1}$ | $10.8_{\pm0.6}$ | *32.3* |
| MTL | 56.2 | $21.3_{\pm0.4}$ | $33.9_{\pm1.3}$ | $8.6_{\pm0.5}$ | $5.6_{\pm0.7}$ | *30.3* | 65.4 | $46.1_{\pm0.4}$ | $66.3_{\pm2.0}$ | $9.9_{\pm0.6}$ | $5.1_{\pm0.7}$ | *43.0* | 74.4 | $66.0_{\pm1.0}$ | $84.8_{\pm0.9}$ | $17.2_{\pm1.3}$ | $8.4_{\pm0.5}$ | *54.2* |
| DMT | 55.5 | $3.1_{\pm0.4}$ | $13.7_{\pm1.4}$ | **$10.5_{\pm0.9}$** | **$6.7_{\pm0.4}$** | *24.2* | 65.3 | $41.3_{\pm0.8}$ | $69.5_{\pm2.6}$ | **$12.9_{\pm1.0}$** | **$6.9_{\pm0.5}$** | *43.5* | 77.9 | $55.6_{\pm1.2}$ | **$89.1_{\pm1.4}$** | **$18.1_{\pm0.9}$** | $8.9_{\pm0.4}$ | *54.6* |
| MoS | 56.9 | $28.6_{\pm0.7}$ | $43.2_{\pm1.8}$ | $9.0_{\pm1.0}$ | $5.4_{\pm0.5}$ | *33.3* | 65.1 | $44.5_{\pm0.6}$ | $68.8_{\pm1.0}$ | $10.7_{\pm0.9}$ | $5.9_{\pm0.5}$ | *43.4* | 79.6 | $66.7_{\pm0.9}$ | $84.9_{\pm1.2}$ | $16.8_{\pm1.0}$ | $8.5_{\pm0.5}$ | *56.0* |
| **EVIC** | **57.1** | **$37.8_{\pm0.5}$** | **$56.9_{\pm0.9}$** | $9.8_{\pm0.5}$ | $5.9_{\pm0.7}$ | *★**37.4*** | **65.8** | **$46.3_{\pm0.6}$** | **$69.9_{\pm0.8}$** | $11.6_{\pm0.6}$ | $6.2_{\pm0.8}$ | *★**44.3*** | **79.7** | **$69.1_{\pm0.9}$** | $86.1_{\pm1.3}$ | $17.9_{\pm0.9}$ | **$9.2_{\pm0.3}$** | *★**57.0*** |

**Models.** We conduct supervised fine-tuning with three popular base models—Mistral-7B-v0.3 (Jiang et al., 2023), Llama-3.1-8B (Dubey et al., 2024), and Qwen2.5-14B (Yang et al., 2024a). We use pretrained models instead of instruction-tuned versions to prevent the models from having already seen some samples in our mixed dataset. For more details of the models, see Appendix A.3.

### 4.2. Main Results

**EVIC boosts the multi-domain fine-tuning of LLMs.** We present the performance of different methods on multiple benchmarks in Table 2. Please note that *DSL is not involved in the performance comparison* and only serves as a reference for the upper limit of each ability. From the table, we make the following observations:

- None of the baselines outperform all others across all metrics, regardless of the base model architecture. This confirms the existence of conflicts between abilities and samples from different domains.

- Our proposed EVIC outperforms all baseline methods in terms of average performance across all metrics. Specifically, EVIC exceeds the best-performing baseline by four points on Mistral-7B and around one point on Qwen2.5-14B. Furthermore, EVIC achieves either the highest or the second-highest results in the individual metrics for each domain.

- Although simple and straightforward, MTL sometimes achieves decent performance. Specifically, its average performance on Mistral-7B surpasses that of DMT, and its accuracy for the mathematical reasoning and Pass@1 for coding on Llama-3.1-8B exceed those of both DMT and MoS. Nevertheless, MTL remains overall inferior to EVIC as well as DMT and MoS, indicating the necessity of data management or curriculum learning in multi-domain datasets.

- DMT exhibits overall stronger general capabilities than EVIC and MoS, which may be attributed to its focus on the general data in the final training stage. However, DMT may lead to unsatisfactory coding performance, especially when code data is scarce (in our experiments, the amount of code data is one-fifth of the math data and less than half of the general data). This is because the small portion of code data added in the final stage cannot fully counteract the decline in coding ability caused by improvements in other capabilities.

- MoS exhibits stronger coding ability than DMT, possibly because it can adaptively adjust data sample probabilities to handle imbalanced domain distributions. However, it still underperforms EVIC. We speculate that an important reason for this performance gap is that the heuristic reward signals in MoS do not directly relate to the model's optimization process, as the interactions between samples do.

**EVIC yields a higher performance-to-sample ratio.** Table 4 presents the number of training steps taken by different methods to achieve the performance shown in Table 2, with a batch size of 128. For EVIC, we also report the number of training steps per iteration. As can be seen from the table, given the same batch size, EVIC achieves higher performance with fewer training steps compared to all the baselines. Specifically, the performance-to-sample ratio of EVIC is at least 1.29 times that of MTL, that is, EVIC takes at most 77.7% of the training steps taken by MTL to outperform all baselines across diverse capabilities. Even more remarkably, for Qwen2.5-14B, these two numbers are 2.11 and 47.4%, highlighting the advantage of our approach in fine-tuning large-scale LLMs.

**EVIC covers almost all samples.** Table 3 reports the sam-

*Table 3.* **Sample coverage rate (%) of EVIC across different domains and over the entire dataset in each iteration.**

| Model | Mistral-7B-v0.3 | | | | Llama-3.1-8B | | | | Qwen2.5-14B | | | |
|---|---|---|---|---|---|---|---|---|---|---|---|---|
| Domain | Code | Math | General | Total | Code | Math | General | Total | Code | Math | General | Total |
| Iteration 1 | 68.99 | 45.37 | 40.27 | 46.51 | 83.42 | 86.09 | 71.75 | 81.70 | 80.73 | 72.62 | 55.85 | 68.72 |
| Iteration 2 | 93.58 | 90.75 | 92.08 | 91.44 | 87.59 | 94.52 | 80.34 | 89.71 | **97.78** | **97.40** | **99.10** | **97.92** |
| Iteration 3 | 97.49 | 96.11 | 95.78 | 96.17 | **99.86** | **99.96** | **99.87** | **99.92** | – | – | – | – |
| Iteration 4 | **98.97** | **99.07** | **98.50** | **98.90** | – | – | – | – | – | – | – | – |

*Table 4.* **The number of training steps taken by different methods to achieve the performance in Table 2.** $|D_{\text{warm-up}}|$ and $|D_{\text{EVIC-iter}m}|$ respectively represent the number of warm-up steps and the number of steps in the $m$-th iteration. $|D_{\text{EVIC}}|$, $|D_{\text{DMT}}|$, $|D_{\text{MoS}}|$, and $|D_{\text{MTL}}|$ respectively represent the number of steps for EVIC (in total), DMT (3 epochs), MoS (3 epochs), and MTL (3 epochs). We use a batch size of 128 for EVIC and all baselines.

| Model | Mistral | Llama3.1 | Qwen2.5 |
|---|---|---|---|
| $|D_{\text{warm-up}}|$ | 72 | 72 | 72 |
| $|D_{\text{EVIC-iter1}}|$ | 661 | 1162 | 978 |
| $|D_{\text{EVIC-iter2}}|$ | 873 | 959 | 973 |
| $|D_{\text{EVIC-iter3}}|$ | 654 | 1010 | – |
| $|D_{\text{EVIC-iter4}}|$ | 1,058 | – | – |
| $|D_{\text{MTL}}|$ (for 3 epochs) | 4,269 | 4,269 | 4,269 |
| $|D_{\text{DMT}}|$ (for 3 epochs) | >4,269 | >4,269 | >4,269 |
| $|D_{\text{MoS}}|$ (for 3 epochs) | 4,269 | 4,269 | 4,269 |
| $|D_{\text{EVIC}}|$ (in total) | 3,318 | 3,203 | 2,023 |
| $|D_{\text{MTL}}|/|D_{\text{DMT}}|$ | <1.00 | <1.00 | <1.00 |
| $|D_{\text{MTL}}|/|D_{\text{MoS}}|$ | 1.00 | 1.00 | 1.00 |
| $|D_{\text{MTL}}|/|D_{\text{EVIC}}|$ | ★**1.29** | ★**1.33** | ★**2.11** |

*Table 5.* **Performance of Mistral-7B trained with the non-iterative variant of EVIC at each iteration.** $\text{EVIC}_{\text{iter}}$ and $\text{EVIC}_{\text{non-iter}m}$ represent the standard version of EVIC and the non-iterative variant for $m$ iterations, respectively.

| $D_{\text{test}}$ | GSM8K | HumanEval | | AlpacaEval 2 | | |
|---|---|---|---|---|---|---|
| Metric | ACC | P@1 | P@10 | LC-WR | WR | *AVG* |
| $\text{EVIC}_{\text{non-iter1}}$ | 51.7 | 18.5 | 30.5 | 8.0 | 4.5 | *27.5* |
| $\text{EVIC}_{\text{non-iter2}}$ | 50.7 | 28.4 | 47.6 | 8.9 | 5.1 | *31.9* |
| $\text{EVIC}_{\text{non-iter3}}$ | 54.3 | 28.0 | 46.3 | 10.1 | 6.1 | *33.2* |
| $\text{EVIC}_{\text{non-iter4}}$ | 52.2 | 27.2 | 48.2 | **12.0** | **6.8** | *33.1* |
| $\text{EVIC}_{\text{non-iter5}}$ | 52.8 | 26.3 | 45.7 | 11.2 | 6.4 | *32.5* |
| $\text{EVIC}_{\text{iter}}$ | **57.1** | **37.8** | **56.9** | 9.8 | 5.9 | *★37.4* |

*Table 6.* **Performance of Llama-3.1-8B trained with the non-iterative variant of EVIC at each iteration.** $\text{EVIC}_{\text{iter}}$ and $\text{EVIC}_{\text{non-iter}m}$ represent the standard version of EVIC and the non-iterative variant for $m$ iterations, respectively.

| $D_{\text{test}}$ | GSM8K | HumanEval | | AlpacaEval 2 | | |
|---|---|---|---|---|---|---|
| Metric | ACC | P@1 | P@10 | LC-WR | WR | *AVG* |
| $\text{EVIC}_{\text{non-iter1}}$ | 62.2 | 43.1 | 68.3 | 10.5 | 5.3 | *41.9* |
| $\text{EVIC}_{\text{non-iter2}}$ | 63.4 | 45.2 | 65.2 | 10.6 | 5.1 | *42.2* |
| $\text{EVIC}_{\text{non-iter3}}$ | 64.1 | 45.1 | 68.9 | 11.4 | 5.4 | *43.2* |
| $\text{EVIC}_{\text{iter}}$ | **65.8** | **46.3** | **69.9** | **11.6** | **6.2** | *★44.3* |

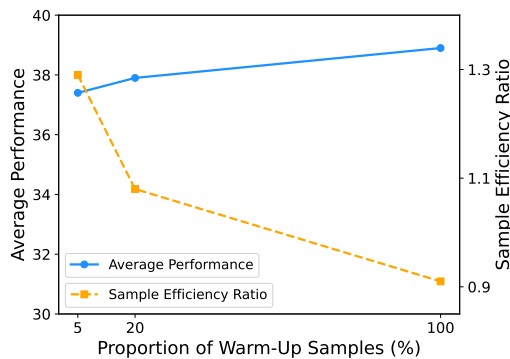

*Figure 3.* Average performance across all benchmarks of Mistral-7B trained with EVIC with different warm-up sample proportions (5%, 20%, and 100%). We default to a 5% proportion for the warm-up data in our main experiments. The "sample efficiency ratio" refers to $|D_{\text{MTL}}/D_{\text{EVIC}}|$ (defined in Table 4).

ple coverage rate of EVIC across domains in each iteration, defined as the proportion of samples that have been learned at least once, excluding the warm-up stage. From the table we find that, although the EVIC framework does not enforce every sample to be selected at least once, nearly 90% of the samples are learned within two iterations, and this proportion exceeds 96% within three iterations. Considering that EVIC selects a sample for training only when it

positively influences the overall dataset, we can conclude that nearly all samples are presented to the model at an appropriate time. This iterative adaptation of the training curriculum, which prioritizes samples that contribute most to the model training, effectively enhances the model's diverse capabilities (as shown in Table 2) and significantly improves performance-to-sample ratio (as shown in Table 4). For an intuitive explanation of this feature, please see the discussion in Section 3.3.

### 4.3. Ablations

**Iterative interaction computation is necessary.** We conduct ablations with Mistral-7B and Llama-3.1-8B to verify

the necessity of the iterative interaction computation in the EVIC framework. In the experiments, we compare EVIC with its non-iterative variant, which only selects samples after the warm-up phase for multiple epochs of learning, without updating the iteration matrix periodically during the training process. Specifically, as shown in Table 3, the Mistral-7B and Llama-3.1-8B models trained with the non-iterative variant are exposed to approximately 46% and 81% of all the samples, respectively. To align their total training steps with those of the standard version of EVIC shown in Table 4, we train Mistral-7B and Llama-3.1-8B using the non-iterative variant for five and three epochs, respectively.

The results are presented in Tables 5 and 6. As can be seen, the performance of the non-iterative variant improves as training progresses. However, despite with similar training steps, its results remain significantly lower than those of the standard version of EVIC. In addition, we find that the performance of the non-iterative variant for Mistral-7B begins to decline after the fourth iteration. Although its performance on AlpacaEval 2.0 exceeds that of the standard version at the fourth iteration, its average performance throughout the training process is at least four points lower than that of the standard EVIC. Thus, we can conclude that: (1) The iterative computation of interactions between samples and the corresponding sample selection are necessary for the effectiveness of EVIC. (2) The effective use of as many samples as possible contributes to the improvement of model performance, which further highlights the importance of EVIC's superiority in the performance-to-sample ratio.

**5% of the samples for warm-up is sufficient.** In the main experiments, we default to using 5% of all samples as warm-up data to help the model adapt to the transition from the pretraining distribution to the SFT distribution. To evaluate the effect of this proportion, we conduct an ablation study with Mistral-7B-v0.3, as shown in Figure 3. Specifically, we report the average performance of the model across all benchmarks and $|D_{\mathrm{MTL}}|/|D_{\mathrm{EVIC}}|$ (defined in Table 4) with the proportions of 5%, 20%, and 100%. As shown in the figure, increasing the proportion of warm-up samples leads to a slight performance improvement (less than 2%), but at the cost of a significant decrease in the performance-to-sample ratio. Therefore, a 5% warm-up sample proportion is sufficient for EVIC to achieve satisfactory performance while maintaining a high performance-to-sample ratio.

## 5. Related Work

Despite the impressive performance of LLMs in specialized domains, research on training them on multi-domain datasets remains in its early stages (Wang et al., 2023a; Leybzon & Kervadec, 2024; Mueller et al., 2022; Chen et al., 2025; 2024). A key challenge stems in the conflicts between samples from different sources and domains (Su et al., 2024;

Xu et al., 2024b; Wang et al., 2023b; Ge et al., 2023).

Recently, some pioneering studies have explored this problem. Dong et al. (2023) conduct extensive experiments and propose an empirical guideline for fine-tuning LLMs on mixed datasets that include math, code, and general samples: to train on the full amount of math and code data, and then train on the general data with a small proportion of math and code samples included. This may work for some model architectures or datasets, but the lack of theoretical supports limits its reliability. Wu et al. (2024) use reinforcement learning to adjust focus dynamically on various sub-datasets based on their current learning process. Others apply curriculum learning, using prior criteria such as prompt length, attention scores, and loss values to rank sample difficulty and design a learning path from simple to complex tasks (Kim & Lee, 2024), but they do not consider conflicts between samples and are therefore not applicable to multi-domain mixed datasets. In contrast to the aforementioned methods that rely on empirical investigation or heuristic strategies, we move beyond empirical practices by modeling inter-sample interactions between samples as their influence on each other's loss, estimated using gradients. Our approach, which delves deeper into the model training process, allows for more refined management of training samples and provides a reliable training curriculum, thereby improving the model's multi-domain performance and enhancing performance-to-sample ratio.

Another line of research focuses on goal-directed fine-tuning, which aims to select data from multi-domain datasets relevant to specific capabilities, rather than enhancing the model's overall abilities. (Cao et al., 2023; Wu et al., 2024; Zhang et al., 2023; Xia et al., 2024b). Xia et al. (2024b) selects samples for training by predicting the similarity between training and test samples. In addition, some methods rely solely on instruction information to identify relevant tasks (Lee et al., 2024), while others propose a self-guided approach where the student LLM generates task-specific input-output pairs and fine-tunes itself in multiple stages (Zhao et al., 2024). In contrast to the goal of the aforementioned works—leveraging mixed datasets to enhance the model's specialized abilities, we focus on improving the comprehensive multi-domain capabilities of LLMs. Furthermore, from a technical perspective, we consider the interactions between training samples, which the aforementioned works do not take into consideration.

## 6. Conclusion and Future Work

In this paper, we propose EVIC, a novel approach to multi-domain fine-tuning of large language models, which addresses the notorious trade-off between domain-specific abilities. By modeling the evolving interactions between training samples, we go beyond heuristic or empirical meth-

ods, providing a deeper understanding of how these interactions influence the model's learning process. Our approach not only improves performance-to-sample ratio but also mitigates domain conflicts, resulting in significant enhancements across various capabilities. Extensive experiments on a mixed dataset covering math, coding, and general instruction-following tasks, using Mistral-7B, Llama-3.1-8B, and Qwen2.5-14B, demonstrate that EVIC outperforms all baseline methods, highlighting its effectiveness in enhancing multi-domain performance.

Our future work will further explore the theoretical explanations behind EVIC, including the dynamics driving its effectiveness in improving multi-domain capabilities, the theoretical foundations of its superior performance-to-sample ratio, and why EVIC can nearly cover all samples without imposing the constraint that every sample must be learned.

## Acknowledgments

This work was supported in part by National Key R&D Program of China under contract 2022ZD0119801, National Nature Science Foundations of China grants U23A20388 and 62021001. This work was supported in part by Huawei as well. We would like to thank all the anonymous reviewers for their insightful comments.

## Impact Statement

This paper presents work whose goal is to advance the field of Machine Learning. There are many potential societal consequences of our work, none which we feel must be specifically highlighted here.

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

# A. More Details about Experiments

## A.1. Tasks and Datasets

To enhance the capabilities of large models in code generation, mathematical reasoning, and instruction following, we use three distinct datasets (i.e., CodeAlpaca, GSM8k-RFT, and GPT4-Alpaca) to train our models in these respective domains.

- CodeAlpaca[2] (Chaudhary, 2023) is designed for training code generation models and contains 175,000 programming challenges in Python. Each challenge tests model's ability to generate syntactically correct and meaningful code, ranging from simple algorithms to more complex tasks.

- GSM8k-RFT[3] (Cobbe et al., 2021) is employed to evaluate mathematical reasoning, containing over 8,000 arithmetic and algebraic problems that require multi-step reasoning. This dataset tests model's ability to solve complex numerical tasks including problem solving and logical deduction.

- GPT4-Alpaca[4] (Peng et al., 2023) is utilized for instruction following and consists of 1,750 human-written tasks across multiple domains. Model is required to generate correct responses based on diverse instructions, assessing its capacity to understand and execute complex instructions.

## A.2. Baselines, Hyperparameters, and Model Training

**Baselines.** We compare our proposed EVIC framework with the following baselines.

- **Multi-Task Learning** (**MTL**) trains on the mixed dataset without any sample management strategies.

- **Domain-Specific Learning** (**DSL**) trains only on the separate dataset for each domain, which reflects the upper limit of the model's specialized ability on each dataset. We use $DSL_{code}$, $DSL_{math}$, and $DSL_{gen}$ to represent the versions of DSL training only on code, math, and general samples, respectively.

- **Dual-stage Mixed Fine-tuning** (**DMT**, Dong et al. (2023)) trains on full amounts of specialized (i.e., code and math) samples first and then trains on general-domain samples with a proportion of $k$ of the specialized samples included. Dong et al. (2023) recommend $k = 1/256$ in their paper, but we conduct experiments using $k = 1/64, 1/128, 1/256$ to search for an optimal $k$ and thus explore its performance limits.

- **Mixture-of-Skills** (**MoS**, Wu et al. (2024)) employs reinforcement learning with reward signals based on heuristic metrics such as transferability and difficulty to train a scorer network, and then use it to assign sampling probabilities to samples. We run MoS with both of the metrics to explore the upper limit of performance.

**Hyperparameters.** We use a learning rate of $2 \times 10^{-5}$, the cosine learning rate scheduler, a batch size of 128 for all the methods. For LoRA, we use a rank of 128, $\alpha = 512$, a dropout ratio of 0.1, and learn LoRA parameters for all attention matrices.

**Model Training.** We conduct all the model training with LoRA (Hu et al., 2021) implemented in the LLaMA-Factory framework (Zheng et al., 2024). We run DSL until its test performance no longer improves. For MTL, DMT, and MoS, we align their training process to three epochs, as their performance shows only marginal gains beyond three epochs. Because EVIC selects different samples at each stage, we measure its training duration in terms of *iterations*[5] and *training steps* (with a batch size of 128), and ensure that **the total training steps of EVIC do not exceed those of the baseline methods** to maintain firness. We run all experiments on 8 NVIDIA A100 GPUs (80GB).

---

[2]https://github.com/sahil280114/codealpaca
[3]https://huggingface.co/datasets/openai/gsm8k
[4]https://github.com/Instruction-Tuning-with-GPT-4/GPT-4-LLM
[5]An iteration refers to performing one sample selection phase and training the model on the selected samples for one epoch.

## A.3. Models

We conduct supervised fine-tuning with three popular base models—Mistral-7B-v0.3[6] (Jiang et al., 2023), Llama-3.1-8B[7] (Dubey et al., 2024), and Qwen2.5-14B[8] (Yang et al., 2024a). We use pretrained models instead of instruction-tuned versions to prevent the models from having already seen some of the samples in our mixed dataset.

- Mistral-7B-v0.3 is an open-source language model developed by Mistral AI. With 7 billion parameters, it excels in performance, and is designed for efficiency and versatility.

- Llama-3.1-8B is a large-scale language model by Meta, featuring 8 billion parameters. It excels in tasks such as language understanding, generation, and translation across various domains.

- Qwen2.5-14B is a robust language model developed by Qwen AI, offering 14 billion parameters. It focuses on enhancing generative tasks, including content creation and problem-solving, delivering highly accurate and context-aware outputs.

## A.4. Evaluation

For models trained on CodeAlpaca, GSM8k-RFT and GPT4-Alpaca, we evaluate their performance on the HumanEval, GSM8k-test and AlpacaEval benchmarks.

- HumanEval (Chen et al., 2021) consists of 164 programming problems that cover a range of common tasks in the Python programming language. Each problem requires the model to generate a valid Python code solution based on a given problem description. The difficulty of the problems ranges from basic algorithmic tasks to moderately complex programming challenges. We perform model inference with a temperature of 0.3 and random seeds ranging from 1 to 10, and report the mean and standard deviation of Pass@1 and Pass@10 of all models.

- GSM8K (Cobbe et al., 2021) test set is a subset of the GSM8k dataset, consisting of more than 8,000 mathematical problems. These problems are designed to evaluate models' math reasoning abilities, particularly in solving arithmetic and algebraic tasks that require multi-step reasoning. We employ greedy decoding with a temperature of 0.0 and report the accuracies to evaluate models with GSM8k test set.

- AlpacaEval (Li et al., 2023c) consists of 800 instruction tasks across various domains, such as information retrieval, task execution, reasoning, and classification. Each instruction is human-written, and the model's task is to generate the correct response based on it. The dataset tests the model's ability to understand and follow diverse instructions. We perform model inference with a temperature of 0.7 and random seeds ranging from 1 to 10. We employ GPT-4–the default evaluator of AlpacaEval–as the judge to compare the outputs of all models with those of GPT-4, using the length-controlled win rate and the win rate as the metrics.

## B. More Experiments

### B.1. Number of times different samples are selected

We report the numbers of times different samples being selected during the training of Llama-3.1-8B and Mistral-7B in Figure 4.

### B.2. More Curriculum Learning Baselines

The baselines in our initial submission (DMT and MoS) belong to progressive CL and balanced CL methods as categorized in (Soviany et al., 2022). Additionally, we add vanilla CL and self-paced CL as baselines. As shown in Tables 7 and 8, EVIC, DMT, and MoS significantly outperform other CL baselines, underscoring the importance of CL methods tailored for multi-domain fine-tuning.

---

[6] https://huggingface.co/mistralai/Mistral-7B-v0.3
[7] https://huggingface.co/meta-llama/Llama-3.1-8B
[8] https://huggingface.co/Qwen/Qwen2.5-14B

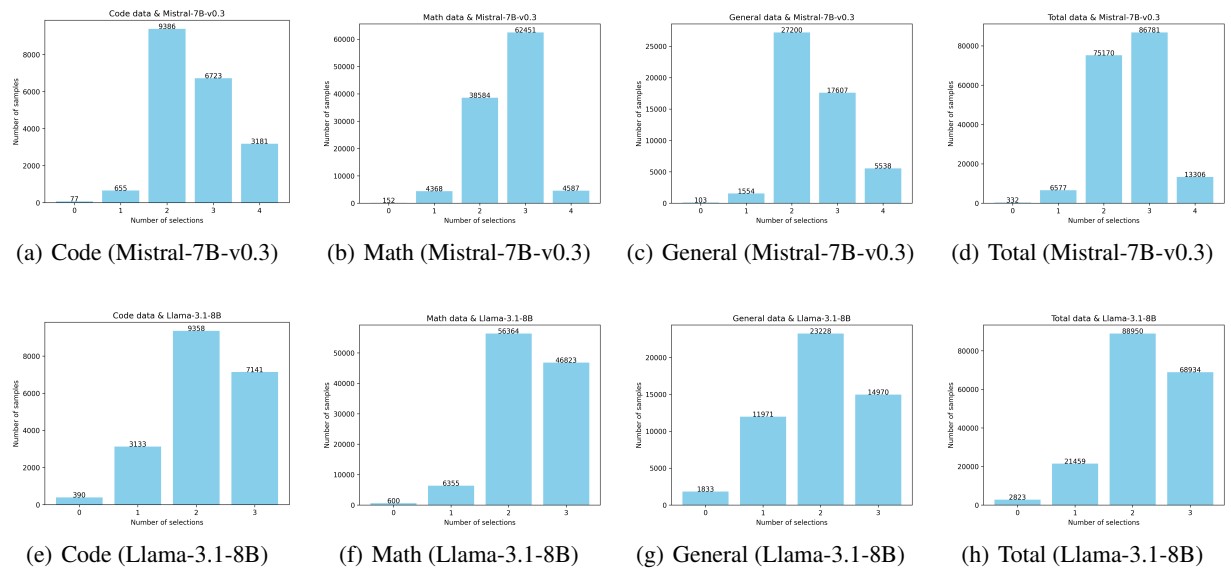

*Figure 4.* Number of times different samples are selected.

*Table 7.* **Performance of Mistral-7B-v0.3 trained on the multi-domain dataset with different methods, evaluated on multiple benchmarks.** Vanilla CL uses query length to measure the difficulty of samples and trains the model in an "easy-to-hard" paradigm accordingly. Self-pased CL measures the difficulty of samples using the model's loss value on the sample and also adopts the "easy-to-hard" manner.

| $D_{\text{test}}$ | GSM8K | HumanEval | | AlpacaEval 2 | | |
|---|---|---|---|---|---|---|
| Metric | ACC | P@1 | P@10 | LC-WR | WR | *AVG* |
| MTL | 56.2 | $21.3_{\pm0.4}$ | $33.9_{\pm1.3}$ | $8.6_{\pm0.5}$ | $5.6_{\pm0.7}$ | *30.3* |
| DMT (progressive CL) | 55.5 | $3.1_{\pm0.4}$ | $13.7_{\pm1.4}$ | $\mathbf{10.5}_{\pm0.9}$ | $\mathbf{6.7}_{\pm0.4}$ | *24.2* |
| MoS (balanced CL) | 56.9 | $28.6_{\pm0.7}$ | $43.2_{\pm1.8}$ | $9.0_{\pm1.0}$ | $5.4_{\pm0.5}$ | *33.3* |
| Vanilla CL | 47.9 | 3.1 | 13.2 | 8.3 | 5.1 | 20.9 |
| Self-paced CL | 48.4 | 3.5 | 11.9 | 7.9 | 4.0 | 20.7 |
| **EVIC** | **57.1** | $\mathbf{37.8}_{\pm0.5}$ | $\mathbf{56.9}_{\pm0.9}$ | $9.8_{\pm0.5}$ | $5.9_{\pm0.7}$ | ★*37.4* |

*Table 8.* **Performance of Llama-3.1-8B trained on the multi-domain dataset with different methods, evaluated on multiple benchmarks.** Vanilla CL uses query length to measure the difficulty of samples and trains the model in an "easy-to-hard" paradigm accordingly. Self-pased CL measures the difficulty of samples using the model's loss value on the sample and also adopts the "easy-to-hard" manner.

| $D_{\text{test}}$ | GSM8K | HumanEval | | AlpacaEval 2 | | |
|---|---|---|---|---|---|---|
| Metric | ACC | P@1 | P@10 | LC-WR | WR | *AVG* |
| MTL | 65.4 | $46.1_{\pm0.4}$ | $66.3_{\pm2.0}$ | $9.9_{\pm0.6}$ | $5.1_{\pm0.7}$ | *43.0* |
| DMT (progressive CL) | 65.3 | $41.3_{\pm0.8}$ | $69.5_{\pm2.6}$ | $\mathbf{12.9}_{\pm1.0}$ | $\mathbf{6.9}_{\pm0.5}$ | *43.5* |
| MoS (balanced CL) | 56.9 | $28.6_{\pm0.7}$ | $43.2_{\pm1.8}$ | $9.0_{\pm1.0}$ | $5.4_{\pm0.5}$ | *33.3* |
| Vanilla CL | 59.3 | 3.4 | 15.9 | 9.6 | 5.8 | 25.6 |
| Self-paced CL | 56.6 | 2.3 | 11.0 | 9.5 | 5.2 | 23.5 |
| **EVIC** | **65.8** | $\mathbf{46.3}_{\pm0.6}$ | $\mathbf{69.9}_{\pm0.8}$ | $11.6_{\pm0.6}$ | $6.2_{\pm0.8}$ | ★*44.3* |

## C. Mathematical Derivations

### C.1. Introduction to Johnson-Lindenstrauss (JL) Transformation

The Johnson-Lindenstrauss (JL) Transformation(Johnson, 1984) is a method for reducing the dimensionality of high-dimensional data while approximately preserving the pairwise distances and inner products. The JL transformation projects data points into a lower-dimensional subspace and ensures that the Euclidean distance between any two points remains the same within a specified error bound.

In practice, to reduce the computational cost on Eq. (1), we employ the JL transformation to project the (Adam) gradients into an 8192-dimensional space, following (Xia et al., 2024a). Specifically , for a given parameter $\theta_t$ and two samples $s^{(i)} = \left(\mathbf{x}^{(i)}, \mathbf{y}^{(i)}\right)$ and $s^{(j)} = \left(\mathbf{x}^{(j)}, \mathbf{y}^{(j)}\right)$, we can compute the $d$-dimensional projections of the gradient of $s^{(i)}$ and the Adam gradient of $s^{(j)}$ by

$$\widehat{\nabla_\theta \ell}\left(\theta_t; \mathbf{x}^{(i)}, \mathbf{y}^{(i)}\right) = \Pi_1^\top \nabla_\theta \ell\left(\theta_t; \mathbf{x}^{(i)}, \mathbf{y}^{(i)}\right), \tag{3}$$

$$\widehat{\Gamma}\left(\theta_t; \mathbf{x}^{(j)}, \mathbf{y}^{(j)}\right) = \Pi_2^\top \Gamma\left(\theta_t; \mathbf{x}^{(j)}, \mathbf{y}^{(j)}\right), \tag{4}$$

with $\Pi_1, \Pi_2 \in \mathbb{R}^{P \times d}$ drawn from a Rademacher distribution. In the equations, $P$ represents the dimension of the gradient, $\widehat{\nabla_\theta \ell}\left(\theta_t; \mathbf{x}^{(i)}, \mathbf{y}^{(i)}\right)$ represents the projected gradient of $s^{(i)}$, and $\widehat{\Gamma}\left(\theta_t; \mathbf{x}^{(j)}, \mathbf{y}^{(j)}\right)$ represents the projected Adam gradient of $s^{(j)}$. In our method, we choose $d = 8192$ following (Xia et al., 2024a).

