# OpenReview forum: "Boosting Multi-Domain Fine-Tuning of Large Language Models through Evolving Interactions between Samples"
_ICML.cc/2025/Conference — ICML 2025 poster_

### Official Review · Reviewer_zLAH · 2025-03-11

**Overall Recommendation:** 3

**Summary:**

The authors propose EVolving Interaction-guided Curriculum (EVIC), a training technique that aims to improve the performance of LLM multi-domain fine-tuning. EVIC iteratively finds the most “helpful” samples in the training set (those that are likely to have helpful influence on the model’s overall loss), then trains on just this helpful subset. The authors conduct experiments of fine-tuning GPT-4 to code, math and general reasoning domains, comparing EVIC to four prior works.

**Claims And Evidence:**

Claims are supported

**Essential References Not Discussed:**

N/A

**Experimental Designs Or Analyses:**

- EVIC performs well in the evaluations compared to prior works (Table 2)
- There are a strong number of baselines and related works included in the evaluation and the evaluation appears to be well-designed by including tasks of varying sizes. However, the evaluation could be expanded and strengthened by including more models beyond GPT-4 and additional domains/datasets.

**Methods And Evaluation Criteria:**

- Lines 159-160 (and other places): when examining the interactions among samples over time, do you find that the interactions eventually stabilize and evolve less? How does that time to stabilization compare with the model convergence time?
- Line 163: How asymmetric are these interactions? Is there any chance that the asymmetry is just due to random noise or do you have any other hypothesis as to why these interactions are asymmetric?
- EVIC begins with a “warm-up” phase, where 5% of samples are randomly selected to start training with. Because 5% is such a small proportion, is the method’s performance highly dependent on which samples are initially selected? Have you tried EVIC with initial portions other than 5%?
- It would be interesting to see a histogram of how often samples are chosen as helpful during training: for instance, a bar for samples never chosen, a bar for samples chosen once, twice, etc.
- If you are training on domains with datasets of varying sizes, will EVIC negatively impact the performance of the model on the rarer domain? For instance, suppose you are training a model to perform n different tasks, where n-1 of the datasets have 10000 samples and the nth dataset has 100 samples. Will the strategy of focusing only on the samples that appear most helpful overall result in neglecting the performance of the smaller dataset?

**Other Comments Or Suggestions:**

Overall, the method is intuitive and offers good performance. However, the analysis and evaluations could be improved by including more tasks and comparing to prior works. These factors together have contributed to my score.

**Other Strengths And Weaknesses:**

One typo: line 252 “firness” instead of “fairness”

**Questions For Authors:**

All questions are asked in the prior sections

**Relation To Broader Scientific Literature:**

Multi-domain training is a useful and important area of ongoing research, and EVIC appears to perform favorably compared to prior works in this area

**Theoretical Claims:**

N/A

---

> ### Author Rebuttal · Authors · 2025-04-01
>
> Dear Reviewer zLAH,
>
> Thank you for your valuable review. We respond to each comment as follows and sincerely hope that our response can properly address your concerns.
>
> Figures and Tables can be found in **zLAH.md** in **https://anonymous.4open.science/r/ICML25-EVIC-D5E8**
>
> # Methods And Evaluation Criteria
>
> > M1: Do you find that the interactions eventually stabilize and evolve less? How does the time to stabilization compare with the model convergence time?
>
> **Res:** We do not observe the stability of the interaction matrix. Taking Llama-3.1-8B and Mistral-7B-v0.3 as examples, the Frobenius norm of the interaction matrix after the warm-up phase, 1 iteration, 2 iterations, and 3 iterations is as follows.
>
> - Llama-3.1-8B: 9e12 -> 5e12 -> 6e12 -> 4e12
> - Mistral-7B-v0.3: 9e12 -> 2e12 -> 4e12 -> 2e12
>
> > M2: How asymmetric are the interactions? Is there any chance that the asymmetry is just due to random noise? Do you have any other hypothesis as to why these interactions are asymmetric?
>
> **Res:** Taking Llama-3.1-8B and Mistral-7B-v0.3 as examples, their interaction matrices consistently have more than **99.7%** of sample pairs $(i,j)$ satisfying $Int(i,j) \neq Int(j,i)$. We further analyze the proportion of pairs $(i,j)$ with opposite influence directions during training, i.e., those satisfying $sign(Int(i,j)) \neq sign(Int(j,i))$. The percentages after the warm-up phase, after 1 iteration, 2 iterations, and 3 iterations are as follows.
>
> - Llama-3.1-8B: 19.3% -> 17.9% -> 16.7% -> 17.4%
> - Mistral-7B-v0.3: 22.1% -> 21.5% -> 21.8% -> 21.1%
>
> The asymmetry of the interaction matrix arises from the difference between each sample's original gradient $Grad$ and Adam gradient $Adam$. Since $Int(j,i) = \langle Adam(j), Grad(i) \rangle$, $Int(i,j) = \langle Adam(i), Grad(j)\rangle$, we naturally have $Int(j,i) \neq Int(i,j)$.
>
> > M3: (1) Have you tried EVIC with initial portions other than 5%? (2) Is EVIC's performance highly dependent on which samples are initially selected?
>
> **Res:** (1) Yes, please see Figure 3 in Section 4.3.
>
> (2) The standard EVIC randomly samples 5% of the dataset for warm-up. To investigate the impact of warm-up samples, we have added additional experiments only using samples from code and general domains for warm-up, denoted as CodeWarmUp and GeneralWarmUp. The results are shown in **Tables zLAH-1 and zLAH-2** in the anonymous link. As shown, unbalanced warm-up sample distribution will reduce model performance.
>
> > M4: It would be interesting to see a histogram of how often samples are chosen as helpful during training. For instance, a bar for samples being never chosen, a bar for samples being chosen once/twice, etc.
>
> **Res:** We have added figures in the anonymous link following your suggestion.
>
> > M5: Will EVIC negatively impact the performance of model on the rarer domain when training on domains with datasets of varing sizes?
>
> **Res:** Imbalanced datasets can indeed hinder the learning of rarer domains. Fine-tuning on such imbalanced datasets is another important challenge in multi-domain finetuning [1,2,3]. To enhance the learning of rare domains, we can weight the interaction matrix computation in EVIC based on the sample size of each domain, increasing the probability of selecting samples from rare domains.
>
> Extending EVIC to imbalanced datasets is a new direction, and we will continue to explore this in future work. If you are interested, we would be glad to discuss this topic further.
>
> # Experimental Designs Or Analyses
>
> > E1 There are a strong number of baselines and related works included in the evaluation and the evaluation appears to be well-designed by including tasks of varying sizes. However, the evaluation could be expanded and strengthened by including more models beyond GPT-4 and additional domains/datasets.
>
> **Res:** Thanks for the positive feedback on our experimental design.
> - **Regarding the models:** Our experiments use three of the most popular and widely-used base models---Mistral-7B-v0.3, Llama-3.1-8B, and Qwen2.5-14B.
> - **Regarding the domains/tasks:** Since DMT is experience-baesd and cannot be directly applied to other domains (as the authors of DMT do not provide relevant experience), we **follow DMT's setting** using mixed dataset containing code, math, and general domains to **ensure fairness**. If you have any other domains or datasets of interest, we would be happy to hear from you through a rebuttal response. **While our computational resources are limited, we will make every effort to conduct additional experiments to follow your suggestions.**
>
> # Other Strengths And Weaknesses
>
> > W1 The "firness" in Line 252 should be "fairness".
>
> **Res:** Thanks for pointing this out. We will correct this typo.
>
> ---
>
> [1] Mixture-of-Skills: Learning to Optimize Data Usage for Fine-Tuning Large Language Models
>
> [2] Massively Multilingual Neural Machine Translation in the Wild: Findings and Challenges
>
> [3] Unsupervised Cross-Lingual Representation Learning at Scale

---

### Official Review · Reviewer_JcLd · 2025-03-14

**Overall Recommendation:** 4

**Summary:**

This paper leverages a way to estimate the training data samples' influence on each other by leveraging the gradients of Adam and projecting it into a lower dimensional space from Xia et al. 2024 [1] and iteratively using this computation in order to select samples to train in the multi-domain fine-tuning setting. They demonstrate that

[1] Xia, Mengzhou, et al. "Less: Selecting influential data for targeted instruction tuning." arXiv preprint arXiv:2402.04333 (2024).

### Update after rebuttal
Given the authors' additional experiments provided, I am inclined to raise my score to 4. I stand corrected in that the authors do show that their method can on average improve scores across different benchmarks. However, I recommend that these two points should be addressed for the updated camera ready/next version.
1. Random seeds should be computed across training runs, not LLM evaluation. Yes I do agree that there is variability in LLM evaluations, but to show consistency in performance improvement, the training run seeds are much more important.
2. The plot of the training iterations/samples vs performance that the authors provided is a great start, although it lacks details about which dataset it was conducted on. Because the novelty of this paper is not the derivation for computing influence of training samples but applying that to multi-domain finetuning, this kind of analysis of how the method is sample efficient compared to other methods seems very important.

**Claims And Evidence:**

The claims are sound.

**Essential References Not Discussed:**

NA

**Experimental Designs Or Analyses:**

Yes, the experimental setup seems sound.

**Methods And Evaluation Criteria:**

The proposed methods and evaluation are sensible for this problem.

**Other Comments Or Suggestions:**

1. A curiosity is whether one can adopt the approach of leveraging test samples to calculate their similarity like in Cao et al. 2023; Wu et al. 2024) in this proposed method to calculate the influence between the test samples vs the training samples and then use the scores to select the training samples. This could verify whether influence computation and sample selection indeed are effective, by leveraging privileged information about the test samples.
2. How are the standard deviations in Table 2 calculated? How many runs of the experiments were done?
3. How expensive is computing the influence at each iteration of EVIC?

**Other Strengths And Weaknesses:**

Strengths
- The paper is clearly written. The problem and the proposed solution (iterative solution to estimate influence and select data samples) are clearly described. A caveat here is that the contribution of the paper isn't to derive the method using Adam gradients to calculate the influence function and it would be poignant to clarify the contributions of this paper vs leveraging insights from other works.
- The experiments are somewhat comprehensive across different benchmarks and models.

Weaknesses
- The results are generally positive, but often times they observe marginal improvement and sometimes no improvement (within the std dev) over other methods at all. For example, in the Qwen and Mistral experiments.
- The sample efficiency is an important point for this method, but Table 3 doesn't necessarily provide a holistic picture of how the evaluation metrics are evolving over the training across the different methods of multi-domain finetuning. It would be very helpful to have that as I mention in the questions below.

**Questions For Authors:**

1. Could you actually provide a more informative version of Table 3 where we can visualize the evaluation metrics over the training (x-axis being the % of total data seen and y-axis the evaluation benchmarks, for example)? It would be important to see the training dynamics and the sample efficiency that way.
2. The authors argue that iterative computation is necessary, but how does the frequency or sample selection size of each iteration change the dynamics of EVIC?

**Relation To Broader Scientific Literature:**

This relates to the language modeling community in investigating how to build models that can excel in different domains when trained altogether.

**Theoretical Claims:**

Yes.

---

> ### Author Rebuttal · Authors · 2025-04-01
>
> Dear Reviewer JcLd,
>
> Thank you for your valuable review. We respond to each comment as follows and sincerely hope that our response can properly address your concerns.
>
> Figures and Tables can be found in **JcLd.md** in **https://anonymous.4open.science/r/ICML25-EVIC-D5E8**
>
> # Other Strengths And Weaknesses
>
> > W1: It would be poignant to clarify the contribution of this paper is not to derive the influence computation using Adam gradients.
>
> **Res:** We will accordingly modify our presentation.
>
> > W2: Often times they observe marginal improvement and sometimes no improvement (within the std dev). For example, in the Qwen and Mistral experiments.
>
> **Res:** We confidently yet humbly believe **EVIC significantly outperforms existing methods** for the following reasons:
>
> 1. Our focus is on the **overall (AVG) performance improvement** in multi-domain fine-tuning. EVIC outperforms the second-best method in terms of AVG by 4.1, 0.8, and 1.0 on Mistral, Llama, and Qwen, respectively. We believe this improvement is **significant**, as **even a 0.5 gain on a multi-domain datasets with inter-sample conflicts is challenging**.
>
> 2. We respectfully speculate that you may find EVIC's improvement in some single domains marginal. However, please note that **EVIC consistently ranks first or second across all domains**, which is also a remarkable and challenging achievement.
>
> > W3: Table 3 does not provide a holistic picture of how the evaluation metrics are evolving over training across different methods of multi-domain finetuning. Could you actually provide a more informative version of Table 3 where we can visualize the evaluation metrics over the training?
>
> **Res:** Due to the deletion of intermediate results and checkpoints for some baselines, we need to retrain them. With limited computational resources, the experiment is still in progress. **We will complete it before the discussion deadline and provide more informative tables or figure as suggested.** We would greatly appreciate your understanding.
>
> # Other Comments Or Suggestions
>
> > S1: Can this method be adopted to calculate the influence between test samples and training samples, and use the scores to select training samples?
>
> **Res:** The approach of calculating the influence between test and training samples is orthogonal to and compatible with our method. However, it faces challenges in multi-domain fine-tuning of LLMs. When test samples conflict (e.g., the evaluation is across different domains), even positively influencing training samples may still conflict with each other, hindering the performance. This brings us back to the core challenge of "how to conduct multi-domain fine-tuning."
>
> > S2: How are the standard deviations in Table 2 calculated? How many runs of the experiments were done?
>
> **Res:** We provide relevant information in Lines 254-264 (left column) of our initial submission but acknowledge it lacks detail. We will revise it as follows.
>
> Specifically, all training is conducted once using the same random seed due to limited resources. For inference:
> - HumanEval: We run model inference with a temperature of 0.3, using random seeds from 1 to 10. Then, we report the mean and standard deviation (std dev) of Pass@1 and Pass@10 using Numpy.
> - GSM8K-test: We use greedy decoding with a temperature of 0.0 (so there is no std dev) and report accuracies.
> - AlpacaEval 2.0: We run inference with a temperature of 0.7, using random seeds from 1 to 10. We use the AlpacaEval library, with GPT-4 as the judge, to compare model outputs against GPT-4 Turbo outputs, and report the mean and std dev of (length-controlled) win rate using Numpy.
>
> > S3: How expensive is computing the influence at each iteration of EVIC?
>
> **Res:** The cost of calculating influence (interactions) in each iteration is mainly dominated by the gradient computation for all samples, which is rougthly equivalent to performing one epoch of MTL. We humbly believe the additional computational cost of the interaction matrix is **acceptable and worthwhile**. For more details, please see our response to **C2 of Reviewer q9CU** due to the rebuttal length constraints.
>
> # Questions For Authors
>
> > Q1: Please see W3.
>
> > Q2: How does the frequency or sample selection size of each iteration change the dynamics of EVIC?
>
> **Res:** The higher the frequency and the fewer samples learned per iteration, the more accurate the gradient computation and interaction matrix estimation become, leading to better EVIC performance. However, excessively high frequency would result in numerous full-dataset gradient computations and extremely high costs. Therefore, in our initial submission, we simply set the frequency and sample selection size as follows.
> - Sample selection size: All samples with a non-negative row sum in the interaction matrix are selected, with no limit on the number.
> - Iteration frequency: After all samples selected in the previous iteration have been learned, the process moves on to the next iteration.

---

> > ### Comment · Reviewer_JcLd · 2025-04-03
> >
> > ## Acknowledged
> >
> > - Thank you for the clarifications to my questions, and I stand corrected that Table 2 shows that EVIC does on average improve upon the existing baselines for multi-domain finetuning. The std dev of the results are computed based on random seeds across evaluations, rather than training, which don't give me the certainty that these are statistically significant differences.
> >
> > ## Questions
> >
> > > The approach of calculating the influence between test and training samples is orthogonal to and compatible with our method. However, it faces challenges in multi-domain fine-tuning of LLMs. When test samples conflict (e.g., the evaluation is across different domains), even positively influencing training samples may still conflict with each other, hindering the performance.
> > >
> >
> > The intention of this suggestion was not to use it in practice for multi-domain finetuning, but rather to study, if given the oracle dataset that we want to transfer on, how effective the method is at choosing the "optimal" set of training datapoints. Depending on the time constraint, I understand if this experiment cannot be done.
> >
> > > Specifically, all training is conducted once using the same random seed due to limited resources.
> >
> > So, from what I read from the paper and the rebuttal, the average and std dev of Table 2 are calculated across random seeds for evaluation. But, usually the variance we care about is more about **random seed across different training runs**. It is a bit misleading in my opinion.
> >
> > > We will complete it before the discussion deadline and provide more informative tables or figure as suggested.
> >
> > Thank you for re-running these experiments and look forward to seeing the how the evaluation metrics evolve over training across different methods (b/c a significant point is that EVIC is more sample-efficient than the others while ultimately improving on the final performance).
> >
> > If this result seems promising, I would be willing to raise my score to 4.

---

> > > ### Author Response · Authors · 2025-04-09
> > >
> > > Dear Reviewer JcLd,
> > >
> > > Thank you again for your insightful comments, constructive suggestions, and positive feedback. We respond to your follow-up questions as follows and sincerely hope that our response has properly addressed your concerns. **If so, we would be profoundly grateful if you could kindly reconsider your score.** Please rest assured that *we hold your expert judgment in the highest regard*, and we sincerely hope that this request *does not cause any inconvenience or disturbance*.
> > >
> > > ---
> > >
> > > > Q1: The intention of this suggestion was not to use it in practice for multi-domain finetuning, but rather to study, if given the oracle dataset that we want to transfer on, how effective the method is at choosing the "optimal" set of training datapoints. Depending on the time constraint, I understand if this experiment cannot be done.
> > >
> > > **Res:** Thank you for your patient explanation and your kind understanding. Despite our best efforts, we are unable to complete this experiment before the deadline of the discussion phase due to limited computational resources. We will continue this experiment and include the results in the revised version of our paper.
> > >
> > > ---
> > >
> > > > Q2: So, from what I read from the paper and the rebuttal, the average and std dev of Table 2 are calculated across random seeds for evaluation. But, usually the variance we care about is more about random seed across different training runs. It is a bit misleading in my opinion.
> > >
> > > **Res:** Thank you for pointing this issue out. We report the standard deviation of the model inference results because the inference of LLMs often involves some randomness. However, we acknowledge that the expression in the initial submission is unclear, and we will make the corresponding modification in the revised version of our paper.
> > >
> > > ---
> > >
> > > > Q3: Thank you for re-running these experiments and look forward to seeing the how the evaluation metrics evolve over training across different methods (b/c a significant point is that EVIC is more sample-efficient than the others while ultimately improving on the final performance). If this result seems promising, I would be willing to raise my score to 4.
> > >
> > > **Res:** Thank you for your encouragement. We provide the results in **https://anonymous.4open.science/r/ICML25-EVIC-D5E8/JcLd-Reply-Rebuttal-Comment.md**. As can be seen, EVIC achieves higher performance with less training steps compared to other methods.

---

### Official Review · Reviewer_q9CU · 2025-03-17

**Overall Recommendation:** 2

**Summary:**

This work presents a curriculum learning method to improve the multi-domain fine-tuning of LLMs. Specifically, the idea is to model the Adam gradient interaction between examples and select the example with the best total benefit on learning other examples. This whole process starts with a warmup stage with around 5% of the data and an iterative process using the aforementioned method. In the evaluation on three different datasets, this method is shown to be better than simple multi-task learning and some other baselines in term of jointly learning all the tasks.

**Claims And Evidence:**

This work presents the EVIC method for multi-domain fine-tuning. The experiments are quite convincing in demonstrating the performance improvement with EVIC and how it outperforms MTL and some other baselines. However, I have two main concerns about the evaluation:

1. There is no curriculum learning baseline in the comparison. I understand that curriculum learning methods are usually not specially designed for multi-domain fine-tuning. However, it is still really important to compare EVIC with some basic curriculum learning methods given how similar these methods are on a high level. This can also serve as ablations to demonstrate the effectiveness of the sample interaction modeling part.

2. The EVIC method seems to be really computationally expensive due to the need to compute the gradient for every example when computing the interaction matrix. This makes scaling this method to large datasets difficult. Additionally, this makes all the questions around sample efficiency questionable. From a practical perspective, the more important value that practitioners care about is how many times you compute the gradient on any example, and I don't see an efficiency advantage on that from EVIC.

**Essential References Not Discussed:**

No essential major references.

**Experimental Designs Or Analyses:**

I can't find how important hyperparameter like M and the total length of each iteration is determined in the paper. These are very important hyperparameters for this method.

**Methods And Evaluation Criteria:**

The method makes sense on a high level, but I wonder why the authors compute sample interactions with "Adam" gradients. Momentum plays a huge role in Adam, and I feel it also heavily influences the calculation in Sec. 3.1. Do the authors have any thoughts or have the authors done analyses on this point? Additionally, lines 159-160 mention that the interaction evolves over the course of the training. How much of this evolution is due to the momentum term?

**Other Comments Or Suggestions:**

1. If the authors can add some qualitative examples of how the models select examples, it will make it more intuitive to understand how this method works.
2. Definitions of LC-WR and WR should be added to the paper when they are first introduced.
3. There are two repetitive references to Xia et al., 2024.

**Other Strengths And Weaknesses:**

n/a

**Questions For Authors:**

See above.

**Relation To Broader Scientific Literature:**

Multi-domain fine-tuning is a practically important but relatively underexplored field. This paper introduces a novel algorithm on this based on prior literature of curriculum learning and sample influence modeling.

**Theoretical Claims:**

No theoretical claims.

---

> ### Author Rebuttal · Authors · 2025-04-01
>
> Dear Reviewer q9CU,
>
> Thank you for your valuable review. We respond to each comment as follows and sincerely hope that our response can properly address your concerns.
>
> Tables can be found in **q9CU.md** in **https://anonymous.4open.science/r/ICML25-EVIC-D5E8**
>
> # Claims And Evidence
>
> > C1: There is no curriculum learning (CL) baseline in the comparison. It is imporatant to compare EVIC with some basic CL methods.
>
> **Res:** Apologies for the lack of clarity in our paper. The baselines in our initial submission (DMT and MoS) belong to **progressive CL** and **balanced CL** methods as categorized in [1,2]. We will refine this in the paper.
>
> Additionally, we have added **vanilla CL** and **self-paced CL** as baselines. Thus, our experiments have covered **four CL method categories**. **Tables q9CU-1 and q9CU-2** in the anonymous link show that EVIC, DMT, and MoS significantly outperform other CL baselines, underscoring the importance of CL methods tailored for multi-domain fine-tuning.
>
> > C2: EVIC seems to be computationally expensive due to the gradient computation for every sample when computing the interaction matrix.
>
> **Res:** We humbly believe the additional computational cost of the interaction matrix is **acceptable and worthwhile**. Additionally, to reduce costs and improve efficiency, we provide some ideas on implementations and methodological improvements.
>
> 1. EVIC’s gradient computation and total cost are **less than twice** that of MTL, which is accpetable and worthwhile for real-world applications. In multi-domain LLM fine-tuning, **performance bottlenecks are more challenging than efficiency bottlenecks**, as extending training time often fails to overcome performance limits. Our proposed EVIC effectively improves the performance via iterative interaction estimation, thereby making a significant contribution.
>
> 2. However, the efficiency and cost concerns you mentioned are also important. For full dataset gradient computation, we parallelize computations across eight GPUs. **We would like to point out that there is a trade-off between efficiency and performance.** To reduce costs, we could use historical gradients of some samples without recomputing them, but this would result in inaccurate interaction estimates and thus decrease training performance. In our initial submission, **we prioritize performance over efficiency**, but in efficiency-critical scenarios, using historical gradients could be explored.
>
> # Methods And Evaluation
>
> > M1: Why computing interactions with "Adam" gradients? Does the momentum heavily influence the calculation in Sec. 3.1? How much of the evolution of interactions is due to the momentum term?
>
> **Res:** Because the interactions are modeled as influences between samples during "training" and LLMs are usually trained with Adam optimizer, the "Adam" gradients naturally appears in the computation of interactions. In a nutshell, our interactions are **defined based on** Adam gradients the corresponding momentum.
>
> We further add experiments to demonstrate the rationale behind computing interactions based on Adam gradients, where we compare EVIC with its variant that computes interactions using the inner product of the original gradients. Please see **Table q9CU-3** in the anonymous link for details due to rebuttal length constraints.
>
> # Experimental Designs Or Analyses
>
> > M2: I can't find how $M$ and the total length of each iteration is determined in this paper.
>
> **Res:** We provide the relevant information in Lines 248-252 (right column) in our initial submission but acknowledge it lacks detail. We will revise it as follows.
>
> Specifically, we set the number of iterations $M$ to be as large as possible while ensuring that the total training steps of EVIC do not exceed those of baselines to maintain fairness. Thus, $M=4,3,2$ for Mistral, Llama, and Qwen experiments, respectively. As for the length of each iteration, it refers to the number of training steps required to learn all selected samples in this iteration, i.e., $|D_m|/{\rm batch size}$.
>
> # Other Comments Or Suggestions
>
> > S1: Adding some qualitative examples of how the models select examples will make it more intuitive to understand how EVIC works.
>
> **Res:** Thanks for your valuable suggestion. We have added some qualitative examples in Figure q9CU-1 in the anonymous link, but we are not sure if this is exactly what you want. **Could you please clarify your suggestion in more details so that we can better follow it?**
>
> > S2: Definitions of LC-WR and WR should be added when first introduced.
>
> **Res:** We will add the definitions based on [3]. Due to the rebuttal length constraints, we cannot provide the definitions here. We would greatly appreciate your understanding.
>
> > S3: Two repetitive references.
>
> **Res:** We will make modifications accordingly.
>
> ---
>
> [1] Curriculum Learning: A Survey.
>
> [2] A Survey on Curriculum Learning.
>
> [3] Length-Controlled AlpacaEval: A Simple Way to Debias Automatic Evaluators.

---

> > ### Comment · Reviewer_q9CU · 2025-04-04
> >
> > Thank you for the detailed responses and the new results! I have two follow-up comments.
> >
> > 1. Can you elaborate on "EVIC’s gradient computation and total cost are less than twice that of MTL"? It's not obvious to me. And despite everything you have said, do you agree that the sample efficiency claims in the original paper is a bit misleading?
> >
> > 2. For "Adding some qualitative examples of how the models select examples will make it more intuitive to understand how EVIC works.", I was mainly hoping to see some actual text examples so that I can have a more intuitive understanding about what example order does EVIC prefers. But you really don't have to include this in this rebuttal. It's not super critical and I'll be happy as long as you can include some examples in the final version.

---

> > > ### Author Response · Authors · 2025-04-09
> > >
> > > Dear Reviewer q9CU,
> > >
> > > Thank you again for your careful review, insightful comments, and constructive suggestions. We respond to your follow-up comments as follows and sincerely hope that our response has properly addressed your concerns. **If so, we would be profoundly grateful if you could kindly reconsider your score.** Please rest assured that *we hold your expert judgment in the highest regard*, and we sincerely hope that this request *does not cause any inconvenience or disturbance*.
> > >
> > > ---
> > >
> > > > C1: (1) Can you elaborate on "EVIC’s gradient computation and total cost are less than twice that of MTL"? It's not obvious to me. (2) And despite everything you have said, do you agree that the sample efficiency claims in the original paper is a bit misleading?
> > >
> > > **Res:** Thank you for your question and constructive feedback.
> > >
> > > 1. This conclusion is empirical, and can be roughly estimated as follows. The additional gradient computation cost of EVIC is approximately equivalent to that of MTL training (as both require the gradients of all samples), and the gradient computation cost for updating model parameters in EVIC is less than that in MTL training, since EVIC uses fewer training steps. Taken together, the total cost of EVIC is therefore less than twice that of MTL ($a \approx c, b < c \Rightarrow a + b \lesssim2c$).
> > >
> > > 2. While this additional cost is acceptable and worthwhile, your insightful comment on *sample efficiency* is **greatly appreciated** and makes us realize that our original claim is unclear. What we intend to convey is that, *given the same amount of training samples, EVIC achieves higher performance in multi-domain LLM fine-tuning, whereas other methods are unable to surpass EVIC even with more training steps (as shown in Table 3)*. In retrospect, this intended meaning is more accurately described as "**higher performance gain per sample**" or "**higher performance-to-sample ratio**". We will **accordingly revise all statements** related to *sample efficiency* in the final version and provide a **clear definition** when the term is first introduced.
> > >
> > >     Thank you again for your thoughtful comments and suggestions.
> > >
> > > > C2: For "Adding some qualitative examples of how the models select examples will make it more intuitive to understand how EVIC works", I was mainly hoping to see some actual text examples so that I can have a more intuitive understanding about what example order does EVIC prefers. But you really don't have to include this in this rebuttal. It's not super critical and I'll be happy as long as you can include some examples in the final version.
> > >
> > > **Res:** Thank you for your patient explanation. We will **accordingly include some examples in the final version** following your suggestions. As a demonstration, the examples we will include will be in the following format:
> > >
> > > - *After warm-up training for Llama-3.1-8B, more than 83% of code and math samples are selected, but only 71.75% of general samples are selected. Which general samples are preferred by EVIC in the beginning of the training stage? We choose three general samples with the largest row sums and three general samples with the smallest row sums from the interaction matrix, as shown below. From their text, it can be observed that...*
> > > - *In the second training iteration of Llama-3.1-8B, the coverage of math samples and general samples increases by 8.43% and 8.59%, respectively. However, the coverage of code samples increases by only 4.17%. Which code samples are newly selected in this stage by EVIC? We choose three code samples that are included in this iteration but not in the first iteration, as shown below. From their text, it can be observed that...*
> > > - *After training for three iterations of Llama-3.1-8B, only 0.08% of the samples have never been selected, meaning they consistently conflict with the majority of samples. We sample three examples from them as follows. From their text, it can be observed that...*
> > > - *More examples...*
> > >
> > > Besides, we will also include some figures to show the frequency with which samples from different domains are selected, as illustrated in **Figure zLAH-1** to **Figure zLAH-8** in https://anonymous.4open.science/r/ICML25-EVIC-D5E8/zLAH.md.

---

### Decision · Program_Chairs · 2025-05-01

**Decision:**

Accept (poster)

**Comment:**

This paper introduces EVIC, a curriculum learning strategy for multi-domain LLM fine-tuning that iteratively selects beneficial training samples based on estimated gradient interactions. The method is well-motivated and addresses the important challenge of balancing performance across diverse tasks. Reviewers generally found the paper clear (JcLd) and the method intuitive (zLAH), acknowledging its novelty (q9CU). The empirical results demonstrate improved average performance across multiple domains and models compared to several baselines, including standard multi-task learning and other curriculum learning approaches (all reviewers, JcLd raised score following rebuttal).

**Strengths:**
* **Novelty and Approach:** Proposes an intuitive and novel iterative method (EVIC) using gradient interactions for multi-domain fine-tuning curriculum (q9CU, zLAH).
* **Improved Average Performance:** Demonstrates consistent average performance gains across diverse domains (code, math, general) and models (Mistral, Llama, Qwen) compared to MTL and relevant CL baselines (q9CU, JcLd, zLAH).
* **Clarity:** The paper is generally well-written, clearly presenting the problem and the proposed EVIC method (JcLd).
* **Reasonable Experimental Setup:** Experiments cover multiple standard models and benchmarks relevant to multi-domain tuning (q9CU, JcLd, zLAH).

**Concerns:**
* **Computational Cost (q9CU, JcLd):** EVIC requires computing gradients for all samples multiple times per epoch to build the interaction matrix, making it significantly more costly than standard fine-tuning.
    * *Address:* Authors argued cost is <2x MTL and justifiable for performance gains, provided training plots showing faster convergence to better results. (q9CU acknowledged explanation; JcLd accepted plots/efficiency argument).
* **Sample Efficiency Framing & Statistical Robustness (q9CU, JcLd):** Initial "sample efficiency" claims needed refinement given the computational cost. Reported standard deviations are based on multiple *evaluation* runs, not multiple *training* runs, limiting conclusions about training stability/robustness.
    * *Address:* Authors promised to rephrase efficiency claims (e.g., "performance gain per sample") and clarify std dev reporting. (q9CU acknowledged; JcLd accepted performance but noted std dev caveat).
* **Contribution Clarity (JcLd):** Initial presentation could better distinguish the paper's contributions versus leveraged methods (Xia et al., 2024).
    * *Address:* Authors promised clarification in the revision. (JcLd acknowledged).
* **Scope and Scalability (zLAH):** Potential challenges with highly imbalanced datasets were noted; evaluation could potentially be broader.
    * *Address:* Authors acknowledged imbalanced data as future work and justified current scope based on resources/baseline fairness. (zLAH acknowledged).
* **Method Details (q9CU, zLAH):** Initial questions regarding Adam gradient rationale, interaction dynamics, and warm-up sensitivity required clarification.
    * *Address:* Authors provided ablations, data, and explanations addressing these points. (q9CU, zLAH acknowledged).

Overall, the paper presents a promising direction for multi-domain fine-tuning with empirical evidence for improvement. While concerns about computational cost and statistical reporting remain, the overall reviewer feedback following the discussion justifies acceptance, conditioned on incorporating the promised revisions.